# Student Grade Evaluation, Survey Feedback, and Lessons Learned during the COVID-19 Pandemic: A Comparative Study of Virtual vs. In-Person Offering of a Freshman-Level General Chemistry II Course in Summer at Xavier University of Louisiana

Navneet Goyal [1,*][ID], Asem I. Abdulahad [1], Janet A. Privett [1], Abha Verma [1], Maryam Foroozesh [1][ID] and Tiera S. Coston [2]

[1] Department of Chemistry, Xavier University of Louisiana, New Orleans, LA 70125, USA; aabdulah@xula.edu (A.I.A.); aprivett@xula.edu (J.A.P.); averma1@xula.edu (A.V.); mforooze@xula.edu (M.F.)
[2] Center for the Advancement of Teaching and Faculty Development, Xavier University of Louisiana, New Orleans, LA 70125, USA; tcoston@xula.edu
* Correspondence: ngoyal@xula.edu

**Abstract:** A primary motivation for this study was to compare student perceptions and performance within a virtual learning environment to the traditional in-person learning experience for the General Chemistry II course taught during a 5-week summer session at Xavier University of Louisiana, a minority serving institution. The authors present quantitative and qualitative analyses including the comparison of student performance on exams during the COVID-19 remote learning experience with exam performance over a 3-year period of conventional in-person instruction. In this article, student grades, survey feedback, and learning outcomes are outlined. This study was performed to assist the faculty in improving and enriching the course content and its delivery, as they coped with the transition to a virtual learning environment imposed by the COVID-19 pandemic.

**Keywords:** history/mission; internet/virtual learning; first-year undergraduates; second-year undergraduates; general chemistry; survey; assessment

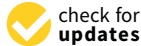



## 1. Background

Xavier University of Louisiana (Xavier) is the only historically Black and Catholic institution of higher education in the United States. Saint Katharine Drexel and the Sisters of the Blessed Sacrament (SBS) sought to create educational institutions from kindergarten through college that serve the African American and America Indian communities. The SBS established elementary schools, a high school, and a normal school, which evolved into Xavier University of Louisiana [1]. In 1925, Xavier University was established in this context to promote stewardship, leadership, and service. Xavier's mission—"The ultimate purpose of the University is to contribute to the promotion of a more just and humane society by preparing its students to assume roles of leadership and service in a global society"—continues to reflect the institution's founding purpose [2]. Xavier is the only Black and Catholic university in the United States. The university continues to produce well-educated graduates positioned to become leaders in the community and to promote Xavier's social justice mission through education, research, and community service. Xavier offers preparation in more than 50 majors on the undergraduate, graduate, and professional degree levels. The university's Fall 2021 enrollment was 3604 (approximately 76.9% African American/Black, 5.0% Asian, 5.7% White, 4.6% Hispanic/Latino, and 7.8% others; approximately 76.0% female, 23.9% male, 0.1% not reported). Of the 2749 undergraduate students, 77.3% majored in Biomedical and Physical Sciences.

According to the U.S. Department of Education, during the past decade, Xavier has ranked first nationally in the number of African American students earning undergraduate



degrees in Biology, Chemistry, Physics, and the Physical Sciences [1]. Xavier also has a national reputation for producing health professionals. In 2012, according to the Association of American Medical Colleges, the university was named the number one undergraduate source of African Americans who complete their medical degrees [1]. In September 2015, the *New York Times Magazine* chronicled the unmatched success of Xavier's premedical program [1]. The iniversity is first in the nation in the number of Black graduates who go on to earn doctorates in the Life Sciences, and fifth in the nation in producing African American students who earn Ph.D.s in Science and Engineering [1]. The College of Pharmacy has also consistently been among the nation's leaders (top 4) in awarding Doctor of Pharmacy degrees to African Americans [1]. The number of students graduating with Chemistry degrees from Xavier is one of the highest in Louisiana. Xavier has been ranked by the American Chemical Society (ACS) as one of the top 25 universities in the nation in awarding bachelor's degrees in Chemistry and has consistently ranked first in the U.S. in producing African American Chemistry graduates [2,3].

## 2. Need for the Study

Noting the magnitude of the COVID-19 pandemic, almost all the academic institutions in the United States and in most other countries shut down their physical campuses to the students, forcing a majority of faculty instructors to urgently adjust to a new virtual learning environment [4–7]. To do so, they have relied on remote teaching delivery platforms provided by Adobe Connect, Google Classroom, Zoom, etc., to reach and serve their students [8–10]. While various institutions have selected different tools, many have focused on addressing and overcoming similar challenges as they had to adapt to emergency remote learning over the more conventional online learning [11–18].

Similar to many small liberal arts colleges, Xavier, a predominantly undergraduate institution (PUI), provides a conducive learning environment that is based on extensive faculty–student interactions and small class sizes. Xavier students have ample access to a host of resources, such as free tutoring, a campus-wide open-door policy for faculty and staff, and peer mentoring. It is important to note that Xavier University did not offer any virtual course offerings in Chemistry before COVID-19.

During the campus shutdown, Xavier was swift in trans-mediating academic resources to a virtual environment. All classes, tutoring, advising, and office hours were conducted synchronously via Zoom from March to August 2020. Since the Fall 2020 semester, following social distancing and other CDC (Centers for Disease Control and Prevention) COVID-19 guidelines, the university has slowly and methodically transitioned some services, including the majority of classes, back to campus. However, because of their large enrollment, General Chemistry I and II courses continued to be taught virtually until Fall 2021 semester.

Many studies have been performed and published on the various approaches taken in higher education to deal with the pandemic-related limitations and their outcomes. Gamage et al. [17] reported new delivery methods and practices for teaching lecture and laboratory courses. Pilkington and Hanif reported how they used technology by providing pre-recorded lecture videos to the students rather than live streaming of lectures [18]. Studies comparing student learning in in-person vs. virtual or hybrid modalities have suggested that there are no significant differences between the different modes of course content delivery. Rather, student performance primarily depends on the pedagogical approaches used to deliver course content [19,20]. Since the beginning of the pandemic, the change from in-person to virtual instruction has been rapid and unprecedented, and many students have found it challenging to adapt. Socioeconomic inequities such as lack of reliable Internet connectivity and access to adequate computer equipment added more uncertainty for a large number of students. Xavier University used a student survey to identify such problems and addressed them to some extent with a laptop loan program and parking lot Wi-Fi zones where the students could connect to the Internet in their cars. Additionally, in both Xavier students and college students around the globe, the

COVID-19 pandemic has induced a variety of negative emotions, including frustration, anxiety, and isolation [21–23]. According to an institutional survey administered soon after the end of the Spring 2020 semester, first-time freshmen overwhelmingly preferred a return to in-person classes for the Fall 2020 semester, while continuing students indicated mixed preferences for virtual and in-person or a combination of the two. As the academic world transitioned to a predominantly virtual space, the authors developed this study to analyze the perceptions of students in the General Chemistry II course and to address the following questions:

1    How easy or challenging was it for the students to adapt to virtual learning?
2    What indicators are important for understanding student adaptation to virtual learning?
3    How can we improve student achievement of learning outcomes in a virtual environment?

### 3. Study Details and Results

The impetus for the study described in this paper was the precipitous decline in student performance in the General Chemistry II course (CHEM 1020) at Xavier. This study specifically focused on student performance in CHEM 1020 within the accelerated summer session course. For the summer session course, this study showed the impact of the COVID-19 pandemic on CHEM 1020 students' grades compared to the previous 3 years of CHEM 1020 instruction. The authors sought to identify the factors that might be responsible for the observed differences and investigate whether they could be countered. The data used in the study were collected at Xavier from 2017 to 2020 for the 5-week summer session course. The traditional General Chemistry courses at Xavier consist of three distinct components: (1) classroom instruction; (2) formative assessment and group learning activities; and (3) summative assessment using multiple-choice exams. In addition, student learning is supported through easy access to peer tutors, open-door office hours, and regularly scheduled review sessions. During the in-person lectures, students are introduced to new concepts through a variety of pedagogical strategies (i.e., traditional lectures, group/collaborative learning, just-in-time teaching, etc.). Formative assessments and group learning activities are accomplished during an instructional period called *drill*. Students are required to enroll in the CHEM 1020 lecture and drill concurrently. The drill sessions begin with short quizzes on pertinent topics and concepts discussed in lecture. Students receive immediate feedback on these quizzes and have a chance to earn a portion of the points lost by working with their peers on drill problems similar to those missed during the quiz. The drill period, a primary source of student engagement and formative assessment during traditional in-person instruction, has traditionally been a cornerstone of Xavier's success in producing graduates in the Physical and Life Sciences [24]. Finally, exams consisting of multiple-choice questions are used for summative assessments. Points earned in lecture and drill components are combined and, based on these, the final course grades in CHEM 1020 are assigned.

Throughout the 5-week summer session, there are 10 formative assessments (drill quizzes) and 4 exams (3 semester exams and a comprehensive final exam). Drill sessions are scheduled 3–4 times per week, for a period of 2 hours each. During this study, the summer class sizes varied from 32 to 48 students. To increase instructor–student interaction opportunities, the lecture students were divided into two drill sections. The abovementioned details are summarized in Table 1.

**Table 1.** Representation of course curriculum.

| Course Components | Length |
| --- | --- |
| Traditional Classroom Instruction: Lecture | 5 Days/Week (each session ~85 min) for 5 weeks |
| Formative Assessment and Group Learning Activities: Drill | Problem-Solving Session and Drill Quizzes (60–90 min/3–4 times per week) |
| Summative Assessment: Exams | 3—Exams (50 min each)<br>1—Final Exam (120 min) |

Prior to the COVID-19 pandemic, the Chemistry Department faculty were not enthusiastic about offering virtual Chemistry courses and had no such plans to do so for the foreseeable future; the mandatory transition to remote learning was the impetus to adapt the CHEM 1020 curriculum classroom instruction and the drill period to the virtual environment. As previously mentioned, student engagement and formative assessment during the traditional in-person drill sessions have played important roles in the success of Xavier graduates, and thus, a change to virtual instruction represented a true dilemma [24]. It was decided to conduct classes synchronously on the Zoom platform, which enabled interaction between the instructor and students, and use a document camera to solve problems and PowerPoint slides to convey information. All the lectures were recorded using Zoom's recording feature and made available to students through Xavier's learning management system (LMS), Brightspace. Formative drill quizzes were administered through the "Quizzes" tool in Brightspace and proctored in real time using Zoom's video conferencing feature. Zoom's "Breakout Rooms" feature was used to encourage peer-to-peer interactions during group learning activities after the drill quizzes. Summative assessments (three semester exams and a comprehensive final exam) were also administered through Brightspace. A key difference between in-person iterations of the course and the synchronous virtual offering during the Summer 2020 session was that the instructors were unable to provide meaningful immediate feedback on drill quizzes administered online. Specifically, online drill quizzes could only be graded as correct or incorrect, whereas in-person instructor feedback normally included analysis of the student's approach to solving each question.

Students enrolled in the Summer 2020 session were asked to complete a voluntary survey designed by course instructors. The survey was administered using Qualtrics software (www.qualtrics.com and accessed on 12 January 2022). A total of 27 out of 47 students, who were at freshmen level (2nd semester), volunteered to participate in the survey, which included the six questions listed in Table 2. The class consisted of 85% female students, which is typical for the institution. The survey responses were anonymous, with no identifying information. The survey was submitted to the Institutional Review Board (IRB) for approval; however, because of time constraints, it was not reviewed prior to administration. The survey was later reviewed by the IRB with no concerns/inquiries.

**Table 2.** Course survey items.

| Question Number | Question |
| --- | --- |
| Q1 | *Have you ever taken a full semester of General Chemistry I or General Chemistry II lecture and drill at Xavier?*<br>*Yes/No* |
| Q2 | *Is this the first time you have taken a full semester of General Chemistry lecture and drill online?*<br>*Yes/No* |
| Q3 | *How would you compare your learning outcomes between online classes and in-person classes?*<br>*(a) No difference*<br>*(b) Nothing to compare*<br>*(c) Online is better*<br>*(d) In-person is better* |
| Q4 | *How do you feel your performance would have differed if you had an in-person lecture/drill course this semester?*<br>*(a) No difference*<br>*(b) Grade would have been better*<br>*(c) Grade would have been worse* |
| Q5 | *Not being at Xavier, what is the most important part of the in-person lecture/drill system that you miss? (Choose all that apply)*<br>*(a) Reinforcement quiz group discussion*<br>*(b) Tutoring center*<br>*(c) One-to-one teacher-student interactions*<br>*(d) Other (please explain)* |
| Q6 | *Is there anything else you would like to share about your experiences in General Chemistry this semester? (optional)* |

The data from the previous three summer sessions (2017–2019) of in-person instruction were compared to that of the virtual Summer 2020 session (Table 3).

**Table 3.** Average grades for summative assessments (three semester exams and final).

| | Summer 2017 | Summer 2018 | Summer 2019 | Average (2017–2019) | Summer 2020 |
|---|---|---|---|---|---|
| Exam 1 | 84.7% | 79.0% | 81.7% | 81.8% | 74.2% |
| Exam 2 | 72.0% | 75.0% | 80.7% | 75.9% | 62.1% |
| Exam 3 | 74.0% | 78.0% | 66.3% | 72.8% | 65.7% |
| Final Exam | 73.7% | 76.0% | 68.5% | 72.3% | 62.9% |

Upon comparing the data from the semester exams, final exams, and final GPAs earned in the course, a significant decline was observed from the previous years (2017–2019) to Summer 2020 (Figure 1). Averages of 81.8, 75.9, and 72.3% were observed for semester exams 1, 2, and 3, respectively, for summer sessions in 2017–2019 of in-person instruction, versus 74.3, 62.0, and 65.6% for those in the Summer 2020 session. For the final exam, an average of 72.3% was observed for summer sessions in 2017–2019 vs. 62.9% in 2020, a significant decline of 10 percentage points. These significant drops in the grades earned during the exams impacted the overall course average, leading to a course GPA of 2.03, which was the lowest when compared with the previous 3 years of summer course offerings (Table 3 and Figure 1). The GPA was determined based on final course grades assigned in CHEM 1020, on a "10-point" scale—that is, 90% = A, 80% = B, 70% = C, and 60% = D. It is important to note that the same instructors taught the courses compared in this study, and the difficulty levels and question types used in the exams were also kept the same. As described previously, the General Chemistry course at Xavier relies heavily on faculty–student and peer-to-peer interactions, as well as remediation and collaborative/group learning activities during the drill period. Based on the significant diminishment in student performance on summative assessment, it was speculated that the emotional impact and isolation that were direct results of the COVID-19 pandemic contributed significantly to the reduction in student performance throughout the Summer 2020 virtual CHEM 1020 course. A similar study by Mahdy [25] noted a similar phenomenon in veterinary medical students who voluntarily reported that virtual learning during the COVID-19 pandemic had a negative impact on their performance. Interestingly, Gonzalez et al. [26] noted in their study that the isolation and confinement resulting from the pandemic improved student performance. Recently, another publication reaffirmed that social interaction plays an important role in student engagement and feeling of belonging, particularly for underrepresented minority groups [27].

The survey results provided some key information:

- Of the students completing the survey, 100% had taken a General Chemistry course at Xavier prior to this class (Q1). Because this is the second in the General Chemistry course sequence at Xavier, these results confirmed that all respondents were continuing Xavier students.
- A total of 81% of the students responded that they had taken an online General Chemistry lecture/drill before, while 19% had not (Q2). Since General Chemistry at Xavier was not offered online prior to the COVID-19 pandemic, these students were enrolled in General Chemistry during the Spring 2020 semester, when all courses were urgently moved online mid-semester.
- A total of 92% of the students stated that they had better learning outcomes when taking in-person courses, while 4% found online instruction better, and the other 4% did not see a difference (Q3, Figure 2).
- In total, 88% of students felt that their grades would have been better if they had been enrolled in the course in person, which was in line with the responses to question 3 (Q4, Figure 2).

- A total of 41% of the students stated that they most missed the one-to-one student–instructor interactions; 32% indicated that they most missed the availability of the in-person tutoring center; 14% most missed the reinforcement quiz discussions with their peers; and the rest chose "other" (Q5, Figure 2). The survey results from Q5 showed that students in General Chemistry at Xavier place significant value on in-person interactions.

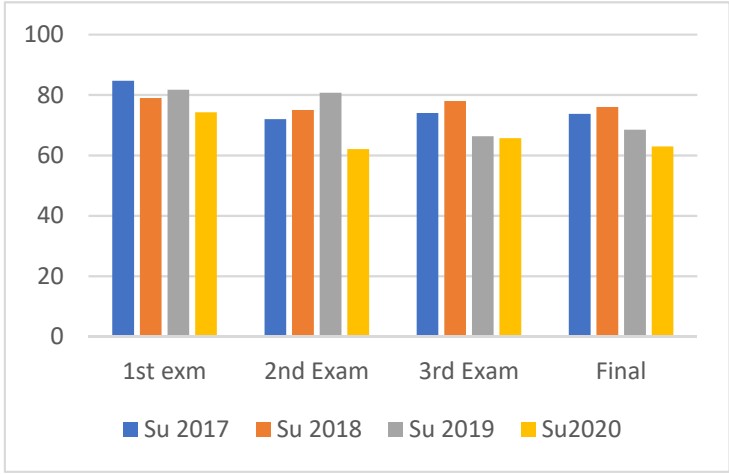

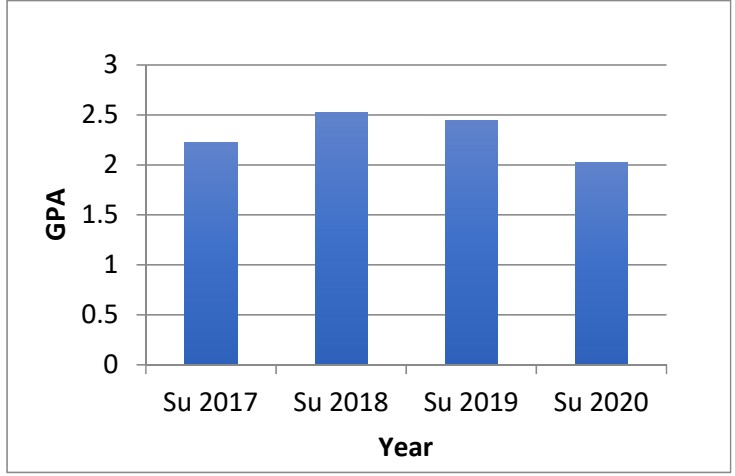

**Figure 1.** Comparison of exam grades and course GPAs from 2017 to 2020.

This information supported the conclusion that the isolation and reduced person-to-person interaction had a negative effect on student performance on summative assessments in the virtual learning environment during the Summer 2020 accelerated session. However, it is also important to note that, prior to the Summer 2020 session, virtual and online instruction for STEM courses at Xavier were taught only in emergency situations such as the shutdown at the beginning of the COVID-19 pandemic during the Spring 2020 semester that immediately preceded the Summer 2020 session. The university responded to this emergency remote learning situation by providing professional development and training opportunities to improve faculty preparedness in online learning. This training was offered during Summer 2020, concurrently with the academic summer sessions. While a lack of faculty preparedness and training in online and virtual instruction may have also contributed to the reduced student performance, the perception of Xavier students in the Summer 2020 General Chemistry II cohort supported the conclusion that the lack of in-person interaction was a significant factor in their performance.

The challenges faced by students in experiencing the lecture/drill in a virtual synchronous (remote) format in the Summer 2020 session were evident from the comments

received in the survey in response to questions 5 and 6. Only a few students responded to question number 6. The comments below appear as they were written by the students (nothing was corrected, added, or removed).

*"It was a lot harder since it was shorter and online."*

*"The course moved EXTREMELY too fast. This course compared to my general chem I course is completely different from each other. It should be no way why the exams covering three modules where one week apart from each other when the in-person course didn't even do this. If the course mirrored the in-person course as we're used to, I as well as the others taking the course grades would have been better."*

*"I feel like the professors did a good job on teaching the material and helping us out when needed. The course is just fast paced, and it can be hard to remember the recently taught last module while trying to focus on the new one at the same time. Maybe more review sessions would help."*

*"Taking the class online has been extremely difficult. In addition to dealing with the stress of the pandemic, taking a science course online has been extremely stressful. I would never take such an important class online ever again if it was my choice."*

*"I struggle very much with chemistry and I started off doing very well with my drills and exams. But, the last two exams I didn't do well on and I think if I would have had extra help from the tutoring center and study groups at school I would have excelled more."*

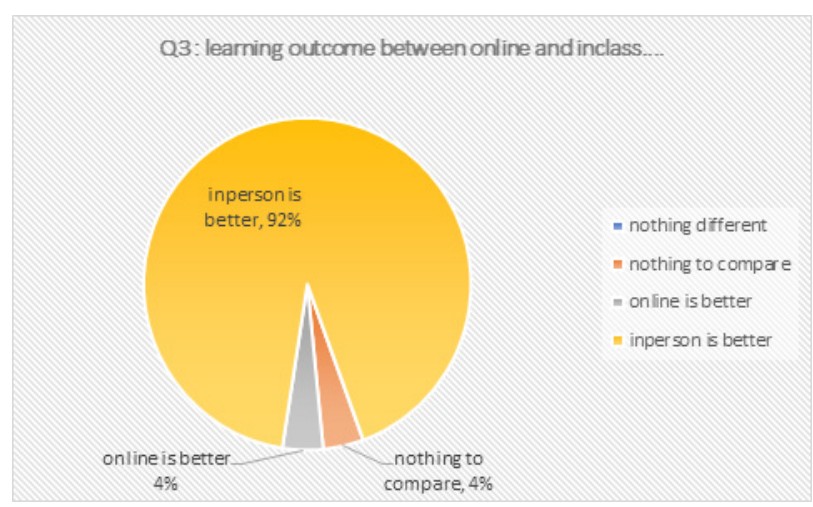

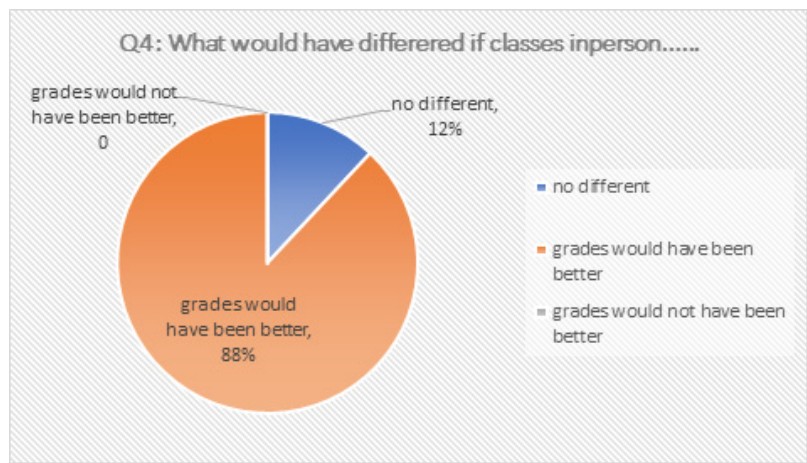

**Figure 2.** *Cont.*

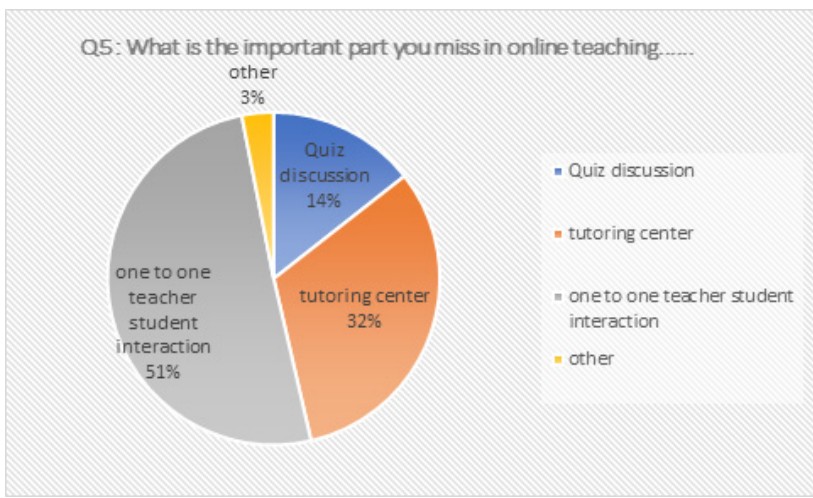

**Figure 2.** Distribution of responses for survey questions 3, 4, and 5.

### 4. Conclusions

The courses included in this study were all taught during the short 5-week summer sessions, and thus were fast paced. The entire CHEM 1020 course was taught virtually for the first time during the 5-week Summer 2020 session. In general, at Xavier, students are advised against taking science courses during these short sessions; however, many use the summer to catch up with their studies. Since the class meets every weekday, keeping up with the material requires discipline and consistency in studying.

Survey data and observations led to the conclusion that students performed below the average in the virtual General Chemistry II course, compared with grades from the same course when offered in-person during the previous three summers. Multiple factors seemed to play a role:

- Xavier students are not accustomed to taking science courses virtually and had difficulty adapting to that learning environment. This was also reflected in a university-wide survey that indicated that the majority of Xavier undergraduates prefer in-person instruction.
- Despite their heavy use during the in-person fall and spring semesters, there was a general lack of student attendance in the free tutoring (group and individual peer-tutoring) and review sessions offered by the Chemistry Resource Center through Zoom. This was further verified in a conversation with the Center's Director, who stated that very few students took advantage of the virtual services offered.
- Students' use of faculty office hours also declined in comparison to in-person sessions. Only a few students attended the Zoom virtual office hours throughout the Summer 2020 session.
- Decreased peer-to-peer interactions in class as well as in study groups affected overall student performance as observed from the survey responses.
- Although synchronous remote instruction using the Zoom platform made virtual teaching more interactive, it was not comparable to the in-person instructor–student interactions, leading to decreased student engagement during class.
- Also as expected, students were dealing with emotional, social, and economic stresses due to the COVID-19 pandemic, which could have affected their academic performance.

It is important to note that in the Summer 2020 session, the university required all faculty to complete intensive training for virtual instruction. The term "virtual" encompasses four different types of instruction at Xavier: (1) Online; (2) Remote; (3) Hybrid; and (4) Mixed Mode. "Online" indicates courses that are taught asynchronously. Students do not have specific virtual class meeting times and may complete work at their own pace within the confines of assignment deadlines. "Remote" refers to courses that are taught

synchronously. In these courses, students have defined times during which they must meet in virtual classrooms to receive instruction. "Hybrid" refers to courses where part of the instruction is in-person and part is virtual. The combination of these two modes of instruction can vary from one course to another, and the virtual portion may be synchronous or asynchronous. "Mixed Mode" denotes courses that are taught in-person (the instructor is physically present in the classroom), but the courses are designed to allow students to attend in-person or virtually and have the same instructional experience no matter how they choose to attend.

Even though most universities shared similar problems, there is also literature available indicating how to introduce distance learning [28]. Research has demonstrated that training to deliver virtual instruction helps instructors improve the quality of the virtual courses they teach and translates into an enhanced overall learning experience for the students enrolled in those courses [29].

Consequently, there arose a need to prepare Xavier's faculty to effectively deliver instruction in any of the virtual modes offered by the university. Training was developed that consisted of three distinct but related parts. The first part, called XULA-Flex, was a 5-week training that was designed to teach the mechanics of Xavier's learning management system (LMS), Brightspace. This training included topics such as creating a gradebook, creating content, and designing and deploying assessments. Once faculty completed XULA-Flex, they were enrolled in a self-paced course called #LearnEverywhereXULA (#LEX). The course was developed based on the eight Quality Matters Course Design Rubric Standards and was designed to demonstrate and model the pedagogical practices and standards that facilitate effective virtual instruction [30]. After completing both XULA-Flex and #LEX, for the third part of their training faculty were required to submit the material developed for a course scheduled to be taught in the Fall 2020 semester for review. The courses were reviewed to observe evidence that the principles and practices demonstrated in XULA-Flex and #LEX were incorporated into the faculty's virtual courses.

In the absence of circumstances such as the current pandemic, the Chemistry Department at Xavier plans to continue offering all Chemistry courses in the in-person format. The faculty firmly believe that the discipline is best taught and learned when students are physically present in the classroom, receive the most personal attention from their instructors, and have opportunities for peer-to-peer interactions. However, if circumstances dictate that Chemistry courses continue to be offered in a virtual format, steps must be taken to address the obstacles identified by students in the Summer 2020 session's virtual CHEM 1020 course. Three trends or themes arose from analysis of the 2017–2020 data. The first trend/theme observed was that students did not use the services of the Chemistry Resource Center or instructor office hours during virtual instruction at the level they did during in-person instruction. The coordinators and instructors of the CHEM 1020 course should incorporate use of the Chemistry Resource Center into the curriculum, such that it would be required for accomplishing certain tasks within the course. Also, instructors should encourage students to visit their virtual office hours by requiring at least one visit each week, even if students do not have specific course-related questions. The single experience of interacting one-to-one with their instructors during virtual office hours may enhance the likelihood of additional visits to seek assistance. The second trend/theme revealed by analysis of the 2017–2020 data was that students missed the peer–peer interactions that helped them to learn course content and prepare for exams. Lack of a sense of community has also been reported in the literature as a weakness in virtual learning [31]. Cox et al. reported a similar result where they showed the significance of the "sense of belongingness" in a large enrollment group in General and Organic Chemistry [32]. It was previously indicated that the Peer-Led Team Learning (PLTL) model plays an important role in enhancing the conceptual understanding of students by reducing their anxiety [33]. To encourage more of this peer–peer interaction in the virtual environment, course instructors and coordinators should facilitate formation of groups where students can work together during and outside of virtual class times to support each other in mastering the course material. Because there

are always students who prefer to and work better on their own, this peer–peer interaction should be strongly encouraged but not required. The third theme/trend observed from this analysis was that teaching CHEM 1020 in a shortened 5-week period exacerbated the pressure experienced by students. While course instructors and coordinators can put a variety of interventions in place as noted above, in order to create a virtual environment that more resembles the in-person environment where students thrive, very little can be done to mitigate the stress induced by a shortened course period in summer sessions. The Chemistry Department will continue discouraging students from taking Chemistry courses in the 5-week summer sessions when possible. However, as some students need to catch up with their studies and progress toward timely graduation, these courses will continue to be offered. In future we will implement new assessment tools (polls, Kahoot, formal surveys, etc.) to evaluate the success and learning outcomes of our students to be disseminated to the education and teaching community [34–36].

**Author Contributions:** Conceptualization, N.G.; methodology, N.G., A.I.A., J.A.P., M.F., T.S.C. and A.V.; formal analysis, N.G., A.I.A., T.S.C., J.A.P. and M.F.; investigation, N.G., A.I.A., J.A.P., M.F., T.S.C. and A.V.; resources N.G., A.I.A., J.A.P., M.F., T.S.C., A.V.; data curation, N.G., J.A.P., A.I.A. and A.V.; writing—original draft preparation, N.G., T.S.C. and A.I.A.; writing—review and editing, N.G., J.A.P., A.I.A., M.F., A.V. and T.S.C.; visualization, N.G.; supervision, N.G.; project administration, N.G.; funding acquisition, N.G. and M.F. All authors have read and agreed to the published version of the manuscript.

**Funding:** This research was Partially funded from the National Institute of General Medical Sciences of the National Institutes of Health under Award Numbers RL5GM118966. The content is the content is solely the responsibility of the authors and does not necessarily represent the official views of the National Institutes of Health.

**Institutional Review Board Statement:** The above named study was presented as a proposal for approval to the Xavier University IRB in the summer of 2020 but was not able to be reviewed until after the data had already been collected. The Xavier University IRB cannot grant retroactive approval to a study. However, this study consists of a simple five-item questionnaire asking students for feedback on a classroom activity. The questions are innocuous and it is made clear to the students that choosing to answer the questions is voluntary. Had this study been reviewed before data collection, it would have been approved as written.

**Informed Consent Statement:** The survey was accompanied by the following statement: The purpose of this short assessment is to gain insight into your experience of taking online and in-person General Chemistry drill and lecture. Your feedback is extremely useful. Completion of this assessment should take no more than five minutes, is anonymous (that is, you do not have to provide any identifying information), and voluntary. You will have the option to leave comments if you wish. Letter is also attached from IRB.

**Data Availability Statement:** Not applicable.

**Acknowledgments:** We acknowledge the initial discussions with Mehnaaz Ali and Ashwith Chilvery. We thank Clair Wilkins-Green for helping and setting up the survey on Qualtrics.com. We also thank all the drill instructors throughout the years for teaching the students and helping in collecting the data. Our thanks also go to all the students who volunteered to participate in survey.

**Conflicts of Interest:** The authors declare no conflict of interest.

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
