# Peer review of "Student Grade Evaluation, Survey Feedback, and Lessons Learned during the COVID-19 Pandemic: A Comparative Study of Virtual vs. In-Person Offering of a Freshman-Level General Chemistry II Course in Summer at Xavier University of Louisiana"

_education, doi:10.3390/educsci12030226_

Round 1

Reviewer 1 Report

Dear authors:

It would be desirable to develop a more extensive review of the research supporting the benefits of students in a virtual learning environment in the fields of General Chemistry.
It is recommended to broaden and deepen the conclusions.

Finally, it is recommended to further explore the usefulness and contribution of the study to the scientific community.

Author Response

Dear Reviewer,

Thank you for the feedback and giving us an opportunity to improve the paper.

There are multiple papers published on COVID-19 impact in higher education. However, this is the only paper in our knowledge, which discusses a fast-paced summer course (5 weeks) during the pandemic. In addition, Xavier is a minority serving institution, has approximately 77% Biological and Physical Sciences majors, and 76% female student body. These aspects provide different perspectives to the study.

As per your advice and recommendation, we performed additional literature searches and included the following papers. Based on these newly added references, we have modified our introduction, discussion, and the conclusion section.

We apologize for potentially missing any important articles.

Please see:

Introduction: Gamage and coworkers have reported new delivery method to teach lecture and laboratory practices.

  1. Gamage, K.A.A.; Wijesuriya, D.I.; Ekanayake, S.Y.; Rennie, A.E.W.; Lambert, C.G.; Gunawardhana, N. Online Delivery of Teaching and Laboratory Practices: Continuity of University Programmes during COVID-19 Pandemic. Educ. Sci. 2020, 10, 291.

Introduction:  Pilkington and Hanif reported how they used technology by providing prerecorded lecture videos to the students rather than live-streaming of lectures.

  1. Pilkington, L.I.; Hanif, M. An Account of Strategies and Innovations for Teaching Chemistry during the COVID-19 Pandemic. Biochem Mol Biol Educ. 2021, 49(3), 320-322. 

Introduction: A new article on Emergency Remote Learning and Online Learning has been added.

  1. Hodges, C.B.; Moore, S.; Lockee, B.B.; Trust, T.; Bond, M.A. The Difference between Emergency Remote Teaching and Online Learning. Available online: https://er.educause.edu/articles/2020/3/the-difference-between-emergency-remote-teaching-andonline-learning (accessed on 2 December 2021).

A reference was added on data collection and survey analysis: Recently another paper is published where it has been reaffirmed that social interaction plays an important role in student engagement and belongingness, particularly for underrepresented racial group.

  1. Thacker, I.; Seyranian, V.; Madva, A.; Duong, N.T.; Beardsley, P. Social Connectedness in Physical Isolation: Online Teaching Practices That Support Under-Represented Undergraduate Students’ Feelings of Belonging and Engagement in STEM. Educ. Sci. 2022, 12, 61.

An article published by Sánchez-Ruiz, L.M  et al. focuses on how B-learning and technology can be an effective tool for effective teaching during Resilience. Our university has implemented similar approaches. A reference has been provided.

  1. Sánchez-Ruiz, L.M.; Moll-López, S.; Moraño-Fernández, J.A.; Llobregat-Gómez, N. B-Learning and Technology: Enablers for University Education Resilience. An Experience Case under COVID-19 in Spain. Sustainability 2021, 13, 3532.

Conclusion section:  Cox et al. have also the significance of sense of belongings in the large enrollment courses of General and Organic Chemistry.

  1. Cox, C.T., Jr.; Stepovich, N.; Bennion, A.; Fauconier, J.; Izquierdo, N. Motivation and Sense of Belonging in the Large Enrollment Introductory General and Organic Chemistry Remote Courses. Educ. Sci. 2021, 11, 549.

Conclusion section:   It has been published previously that Peer-Led Team Learning (PLTL) model can play an important role in enhancing the conceptual understanding of students by reducing the anxiety.

  1. Eren-Sisman, E.N.; Cigdemoglu, C.; Geban, O. The Effect of peer-Led Team Learning on Undergraduate Engineering Students’ Conceptual Understanding, State Anxiety, and Social Anxiety. Chemistry Education Research and Practice 2018, 19, 694-710.

Conclusion statement: In future, we will also implement new assessment tools to evaluate the success and learning outcomes of our students and same can be reported.

  1. Li, K.; Wong, B. The Use of Student Response Systems with Learning Analytics: A Review of Case Studies (2008−2017). International Journal of Mobile Learning and Organization 2020, 14, 63.
  2. Bunce, D. M.; Flens, E. A.; Neiles, K. Y. How Long Can Students Pay Attention in Class? A Study of Student Attention Decline Using Clickers. J. Chem. Educ. 2010, 87, 1438−1443.
  3. Youssef, M. Assessing the use of Kahoot! In an Undergraduate General Chemistry Classroom. J chem Educ. 2022, 99, 1118-1124.

Reviewer 2 Report

This article is well written and provides readers with a sense of how students are faring academically in General Chemistry during the pandemic and perhaps some ways the teaching methods could be changed to enhance learning during forced virtual class meetings.

There have been many recently published articles about student learning/surveys/faculty professional development helping students during the pandemic that should be included in this work. These recent publications will help support the conclusions of the authors as well as echo their findings.

Author Response

Dear Reviewer,

Thank you very much for considering our manuscript for the publication.

As per your advice and recommendation, we performed additional literature searches and included the following papers. Based on these newly added references, we have modified our introduction, discussion, and the conclusion section.

We apologize for potentially missing any important articles.

Please see:

Introduction: Gamage and coworkers have reported new delivery method to teach lecture and laboratory practices.

  1. Gamage, K.A.A.; Wijesuriya, D.I.; Ekanayake, S.Y.; Rennie, A.E.W.; Lambert, C.G.; Gunawardhana, N. Online Delivery of Teaching and Laboratory Practices: Continuity of University Programmes during COVID-19 Pandemic. Educ. Sci. 2020, 10, 291.

Introduction:  Pilkington and Hanif reported how they used technology by providing prerecorded lecture videos to the students rather than live-streaming of lectures.

  1. Pilkington, L.I.; Hanif, M. An Account of Strategies and Innovations for Teaching Chemistry during the COVID-19 Pandemic. Biochem Mol Biol Educ. 2021, 49(3), 320-322. 

Introduction: A new article on Emergency Remote Learning and Online Learning has been added.

  1. Hodges, C.B.; Moore, S.; Lockee, B.B.; Trust, T.; Bond, M.A. The Difference between Emergency Remote Teaching and Online Learning. Available online: https://er.educause.edu/articles/2020/3/the-difference-between-emergency-remote-teaching-andonline-learning (accessed on 2 December 2021).

A reference was added on data collection and survey analysis: Recently another paper is published where it has been reaffirmed that social interaction plays an important role in student engagement and belongingness, particularly for underrepresented racial group.

  1. Thacker, I.; Seyranian, V.; Madva, A.; Duong, N.T.; Beardsley, P. Social Connectedness in Physical Isolation: Online Teaching Practices That Support Under-Represented Undergraduate Students’ Feelings of Belonging and Engagement in STEM. Educ. Sci. 2022, 12, 61.

An article published by Sánchez-Ruiz, L.M  et al. focuses on how B-learning and technology can be an effective tool for effective teaching during Resilience. Our university has implemented similar approaches. A reference has been provided.

  1. Sánchez-Ruiz, L.M.; Moll-López, S.; Moraño-Fernández, J.A.; Llobregat-Gómez, N. B-Learning and Technology: Enablers for University Education Resilience. An Experience Case under COVID-19 in Spain. Sustainability 2021, 13, 3532.

Conclusion section:  Cox et al. have also the significance of sense of belongings in the large enrollment courses of General and Organic Chemistry.

  1. Cox, C.T., Jr.; Stepovich, N.; Bennion, A.; Fauconier, J.; Izquierdo, N. Motivation and Sense of Belonging in the Large Enrollment Introductory General and Organic Chemistry Remote Courses. Educ. Sci. 2021, 11, 549.

Conclusion section:   It has been published previously that Peer-Led Team Learning (PLTL) model can play an important role in enhancing the conceptual understanding of students by reducing the anxiety.

  1. Eren-Sisman, E.N.; Cigdemoglu, C.; Geban, O. The Effect of peer-Led Team Learning on Undergraduate Engineering Students’ Conceptual Understanding, State Anxiety, and Social Anxiety. Chemistry Education Research and Practice 2018, 19, 694-710.

Conclusion statement: In future, we will also implement new assessment tools to evaluate the success and learning outcomes of our students and same can be reported.

  1. Li, K.; Wong, B. The Use of Student Response Systems with Learning Analytics: A Review of Case Studies (2008−2017). International Journal of Mobile Learning and Organization 2020, 14, 63.
  2. Bunce, D. M.; Flens, E. A.; Neiles, K. Y. How Long Can Students Pay Attention in Class? A Study of Student Attention Decline Using Clickers. J. Chem. Educ. 2010, 87, 1438−1443.
  3. Youssef, M. Assessing the use of Kahoot! In an Undergraduate General Chemistry Classroom. J chem Educ. 2022, 99, 1118-1124.

Reviewer 3 Report

General comment: Title of paper “Students Grade Evaluation, Survey Feedback, and Lessons Learned During the COVID-19 Pandemic: A Comparative Study of Virtual vs. In-Person Offering of Freshman-Level General Chemistry II Course in Summer at Xavier University of Louisiana” is too long.

Paper recalls the situation of a Chemistry subject at a particular university during the COVID-19 pandemic.

General feeling is that the paper looks more like a report on an experience than a research paper.  

“Xavier” is cited 41 times within text, plus in the title, abstract and 4 times in the References list, too many, some of them just stating the reputation and well doing of the university as in L20-49, 59-62 or department in L334-341 for instance.

L65-72 look like a news report and are irrelevant for the topic of the paper.

A paper published in an international journal should be of interest for researchers and not look like story-telling board with sensible recommendations at the end on whether some technical course at a given university should be taken in the Summer Session.

Some concrete suggestions:

In the introduction the authors should make clear what is the added value in this paper compared to the existing literature, its aim and the reason why it deserves attention from researchers.

L91-95 lists 3 general questions but it is unclear how and where they are addressed later on. They are not treated as research questions and there is no research methodology on them.

Summer, Spring, Fall should be written consistently in Capital letters.

“course” and “pandemic” should not be capitalized.

L36 Biology, Chemistry, Physics, and the Physical Sciences enumerated degrees should be capitilized.

L36-45 contain several references to [1]  that seem misprints.  

L83-84 The authors transmit a general negative perspective across the globe without deepening and justification. Clearly pandemic was tough but not all institutions reacted in the same way, nor was it their capability to react. The authors should consider more literature with qualitative and quantitative perspective as done in “Sánchez-Ruiz, L.M.; Moll-López, S.; Moraño-Fernández, J.A.; Llobregat-Gómez, N. B-Learning and Technology: Enablers for University Education Resilience. An Experience Case under COVID-19 in Spain. Sustainability 2021, 13, 3532. https://doi.org/10.3390/su13063532” where it seems that some previous experience in using blended methodologies facilitated the transition during the pandemic.

L111-123 should be clearer. Drill sessions seem formative in L111, but in L116-117 it seems that some students can earn some points, but only in Fall and Spring?

L126 cites 48 students, L132 mentions 27 out of 47 students…?

L132-134 do not match with the running discourse of the paragraph.

Q1-Q2 in P4 seem redundant – the authors knew the answer to that question.

Q5 in P5 should have considered “no difference” as possible answer as well.

Percentages in Table 3 should include one tenth for data in each cell.

L184 “Gonzalez and co-workers” should be “Gonzalez et al.”

L317-341 should not be part of the conclusions.

Conclusions should not contain comments on issues that have not been clearly addressed.

In addition to the general and concrete above comments there are some repetitions, unclear sentences or long considerations in the text that have been highlighted and the authors should revise.

Author Response

Dear Reviewer,

Thank you for your comments and suggestions. Please see our detailed responses below.

As this manuscript is intended for international publication, an introduction to the University, it’s values, offerings, accomplishments, and student demographics are important to include.

Comment: In the introduction, the authors should make clear what is the added value in this paper compared to the existing literature, its aim and the reason why it deserves attention from researchers.

Response: Thank you for the comment. There are multiple papers published on COVID-19 impact in higher education. However, this is the only paper in our knowledge, which discusses a fast-paced summer course (5 weeks) during the pandemic. In addition, Xavier is a minority serving institution, has approximately 77% Biological and Physical Sciences majors, and 76% female student body. These aspects provide different perspectives to the study.

Comment: L91-95 lists 3 general questions but it is unclear how and where they are addressed later on. They are not treated as research questions and there is no research methodology on them.

Response: Thank you for the comments: Please see our responses below.

  1. How easy or challenging was it for the students to adapt to virtual learning?

We tried to understand student perspective by doing a survey. We reported students’ experiences based on the responses in lines 210-239, and then in lines 261-281 (comments from the surveys).

  1. What indicators are important for understanding student adaptation to virtual learning?

Please see lines 289-311.

  1. How can we improve student achievement of learning outcomes in a virtual environment?

Please see line 348-387.

Comment: Summer, Spring, Fall should be written consistently in Capital letters.

Response: We have fixed this where appropriate. General use of fall, spring, and summer were not capitalized. Specific ones were capitalized. Thank you.

Comment: “course” and “pandemic” should not be capitalized.

Response: We made changes in the text based on this comment. Thank you.

Comment: L36 Biology, Chemistry, Physics, and the Physical Sciences enumerated degrees should be capitalized.

Response: Changed throughout. Thank you.

Comment: L36-45 contain several references to [1]  that seem misprints.  

Response:  Fixed, thank you.

Comment: L83-84 The authors transmit a general negative perspective across the globe without deepening and justification. Clearly pandemic was tough but not all institutions reacted in the same way, nor was it their capability to react. The authors should consider more literature with qualitative and quantitative perspective as done in “Sánchez-Ruiz, L.M.; Moll-López, S.; Moraño-Fernández, J.A.; Llobregat-Gómez, N. B-Learning and Technology: Enablers for University Education Resilience. An Experience Case under COVID-19 in Spain. Sustainability 2021, 13, 3532. https://doi.org/10.3390/su13063532” where it seems that some previous experience in using blended methodologies facilitated the transition during the pandemic.

Response: Thank you very much for your recommendation and advise. We have mentioned this in our paper, and it also shows how important it is to offer faculty training for such situations. Reference 29 was added in response to this comment.

Comment: L111-123 should be clearer. Drill sessions seem formative in L111, but in L116-117 it seems that some students can earn some points, but only in Fall and Spring?

Response: Thank you. We agree that the above-mentioned section was not completely clear.  It has been clarified.

Comment: L126 cites 48 students, L132 mentions 27 out of 47 students…?

Response: This study was performed during 2017-2020. During this study, the Summer class sizes varied from 32-48 students.  During 2020, class size was 47.

Comment: L132-134 do not match with the running discourse of the paragraph.

Response: This is redundant. This has been removed from the paragraph. Thank you.

Comment: Q1-Q2 in P4 seem redundant – the authors knew the answer to that question.

Response: This is not always true. We often have transfer or transient students who take courses at Xavier during the summer.

Comment: Q5 in P5 should have considered “no difference” as possible answer as well.

Response: Thank you. We will rectify this in future studies.

Comment: Percentages in Table 3 should include one tenth for data in each cell.

Response: Thank you, fixed.

Comment: L184 “Gonzalez and co-workers” should be “Gonzalez et al.”

Response: Thank you, fixed.

Comment: L317-341 should not be part of the conclusions.

Response: We have included the reference provided by you and it now makes sense to include these lines.

Comment: Conclusions should not contain comments on issues that have not been clearly addressed.

In addition to the general and concrete above comments there are some repetitions, unclear sentences or long considerations in the text that have been highlighted and the authors should revise.

Response: As per your advice and recommendation, we performed additional literature searches and included the following papers. Based on these newly added references, we have modified our introduction, discussion, and the conclusion section.

We apologize for potentially missing any important articles.

Please see:

Introduction: Gamage and coworkers have reported new delivery method to teach lecture and laboratory practices.

  1. Gamage, K.A.A.; Wijesuriya, D.I.; Ekanayake, S.Y.; Rennie, A.E.W.; Lambert, C.G.; Gunawardhana, N. Online Delivery of Teaching and Laboratory Practices: Continuity of University Programmes during COVID-19 Pandemic. Educ. Sci. 2020, 10, 291.

Introduction:  Pilkington and Hanif reported how they used technology by providing prerecorded lecture videos to the students rather than live-streaming of lectures.

  1. Pilkington, L.I.; Hanif, M. An Account of Strategies and Innovations for Teaching Chemistry during the COVID-19 Pandemic. Biochem Mol Biol Educ. 2021, 49(3), 320-322. 

Introduction: A new article on Emergency Remote Learning and Online Learning has been added.

  1. Hodges, C.B.; Moore, S.; Lockee, B.B.; Trust, T.; Bond, M.A. The Difference between Emergency Remote Teaching and Online Learning. Available online: https://er.educause.edu/articles/2020/3/the-difference-between-emergency-remote-teaching-andonline-learning (accessed on 2 December 2021).

A reference was added on data collection and survey analysis: Recently another paper is published where it has been reaffirmed that social interaction plays an important role in student engagement and belongingness, particularly for underrepresented racial group.

  1. Thacker, I.; Seyranian, V.; Madva, A.; Duong, N.T.; Beardsley, P. Social Connectedness in Physical Isolation: Online Teaching Practices That Support Under-Represented Undergraduate Students’ Feelings of Belonging and Engagement in STEM. Educ. Sci. 2022, 12, 61.

An article published by Sánchez-Ruiz, L.M  et al. focuses on how B-learning and technology can be an effective tool for effective teaching during Resilience. Our university has implemented similar approaches. A reference has been provided.

  1. Sánchez-Ruiz, L.M.; Moll-López, S.; Moraño-Fernández, J.A.; Llobregat-Gómez, N. B-Learning and Technology: Enablers for University Education Resilience. An Experience Case under COVID-19 in Spain. Sustainability 2021, 13, 3532.

Conclusion section:  Cox et al. have also the significance of sense of belongings in the large enrollment courses of General and Organic Chemistry.

  1. Cox, C.T., Jr.; Stepovich, N.; Bennion, A.; Fauconier, J.; Izquierdo, N. Motivation and Sense of Belonging in the Large Enrollment Introductory General and Organic Chemistry Remote Courses. Educ. Sci. 2021, 11, 549.

Conclusion section:   It has been published previously that Peer-Led Team Learning (PLTL) model can play an important role in enhancing the conceptual understanding of students by reducing the anxiety.

  1. Eren-Sisman, E.N.; Cigdemoglu, C.; Geban, O. The Effect of peer-Led Team Learning on Undergraduate Engineering Students’ Conceptual Understanding, State Anxiety, and Social Anxiety. Chemistry Education Research and Practice 2018, 19, 694-710.

Conclusion statement: In future, we will also implement new assessment tools to evaluate the success and learning outcomes of our students and same can be reported.

  1. Li, K.; Wong, B. The Use of Student Response Systems with Learning Analytics: A Review of Case Studies (2008−2017). International Journal of Mobile Learning and Organization 2020, 14, 63.
  2. Bunce, D. M.; Flens, E. A.; Neiles, K. Y. How Long Can Students Pay Attention in Class? A Study of Student Attention Decline Using Clickers. J. Chem. Educ. 2010, 87, 1438−1443.
  3. Youssef, M. Assessing the use of Kahoot! In an Undergraduate General Chemistry Classroom. J chem Educ. 2022, 99, 1118-1124.

Round 2

Reviewer 1 Report

Accept in present form

Author Response

Dear Reviewer,

Thank you very much for your time and help us in improving the manuscript. We appreciate it.

Best Regards,

Authors

Reviewer 3 Report

Paper is clearer and the authors have fixed as much as they could.

Authors should have in mind that citing reference number should be done just after the original authors or at the end of the sentence, separated by commas, in a parenthesis or nothing, but never behind a period "." that indicates that sentence has ended.

For instance, in L79-82:

"Gamage et al. have reported new delivery methods and practices for teaching lecture and laboratory courses.[18] Pilkington and Hanif reported how they have used technology by providing pre-recorded lecture videos to the students rather than live streaming of lectures.[19]"

might be changed into  

"Gamage et al. [18] have reported new delivery methods and practices for teaching lecture and laboratory courses. Pilkington and Hanif reported how they have used technology by providing pre-recorded lecture videos to the students rather than live streaming of lectures, [19]."

or similarly, but [18] and [19] should not be after the period.

The same applies to all cited references throughout the text,

Author Response

Dear Reviewer,

We appreciate your time, feedback and help to give us an opportunity to improve.

We have fixed all the citations as recommended/ advised by you.

Best Regards,
